# OraclePRM: Unlocking the Potential of Each Instance for Multimodal Process Reward Model Training

## Abstract

Training multimodal process reward models (PRMs) is hard due to (i) distribution shift between training set and test set and (ii) quality imbalance across training data samples. While domain-level reweighting (e.g., DreamPRM) aligns training with test-time objectives, it leaves a clear gap to an oracle upper bound (pass@$N$), even under a "sanity check" that uses test set data to probe headroom—pointing to meta-level under-parameterization. We introduce OraclePRM, an instance-level reweighting framework that assigns an adaptive weight to every training example via bi-level optimization. To realize instance reweighting across scales, we develop two complementary regimes: Instance Table, which learns explicit per-sample weights and excels on small/medium data, and Instance Net, a lightweight neural network that generalizes better and scales to large corpora. A practical, stable training recipe—time-scale matching between upper/lower updates, cold-start initialization, and bounded-range weights—prevents divergence. Integrated with test-time scaling, OraclePRM attains 84.6 accuracy on the MMMU validation set and, when paired with a leading backbone (e.g., GPT-5-mini), achieves first-place results on public multimodal reasoning leaderboards. Moreover, extensive experiments, including benchmark evaluations, baseline comparisons, and a sanity check, demonstrate that OraclePRM closes the gap toward the oracle, achieves leading performance, and trains stably.

## 1 Introduction

Recent advances in reasoning (Snell et al., 2024) have substantially boosted large language models (LLMs) (Brown et al., 2020; Chowdhery et al., 2022; Touvron et al., 2023; Qwen et al., 2025), with Process Reward Models (PRMs) (Lightman et al., 2024; Li et al., 2023) providing step-level supervision and more reliable selection of reasoning trajectories. Extending PRMs to multimodal LLMs (MLLMs) (Wu et al., 2023; Li et al., 2025) is therefore a natural progression. However, multimodal inputs couple high-dimensional visual features with discrete language tokens, enlarging the input space and intensifying *distribution shifts* (Song et al., 2025), while multimodal reasoning corpora suffer pronounced *quality imbalance* (Yu et al., 2024b; Lu et al., 2024) in which noisy or trivial samples dilute effective training (See Appendix Fig. 8). As a result, directly applying text-only PRM methods (Wang et al., 2024f; Luo et al., 2024) yields limited gains and weak generalization in the multimodal setting (Dong et al., 2024).

To address these issues, DreamPRM (Cao et al., 2025) adopts a *domain-reweighting* bi-level optimization scheme. At the lower level, the PRM parameters are optimized under a domain-reweighted training loss, allocating more capacity to high-quality sources and down-weighting noisy ones to correct quality imbalance. At the upper level, a validation objective on a separate meta domain is used to simulate test-time scaling (e.g., aggregation losses aligned with Best-of-$N$ selection), and the resulting signal updates the domain weights per dataset following a meta-learning setup (Shu et al., 2019; Fan et al., 2024b; Sow et al., 2025; Finn et al., 2017), thereby aligning learning with the test-time distribution. In short, the upper level learns domain weights on held-out data that mimic test-time scaling, while the lower level updates PRM parameters under those weights.

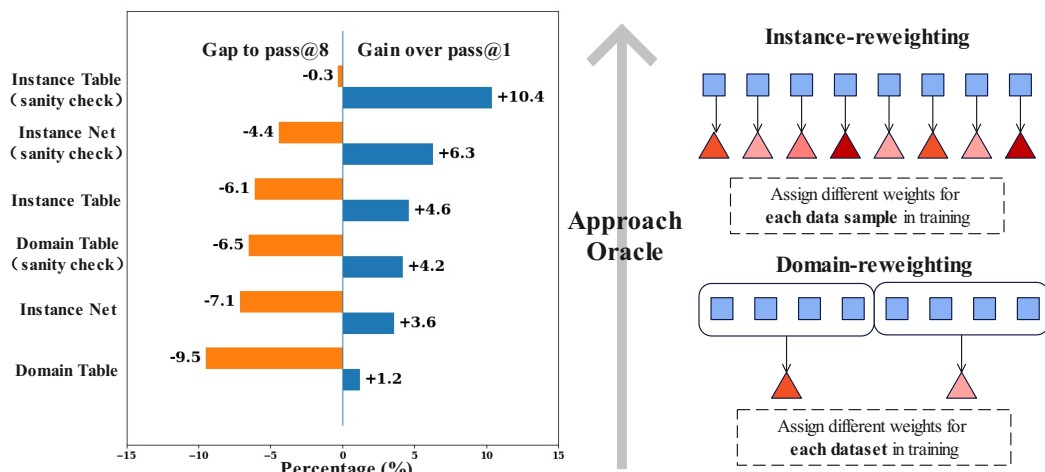

Figure 1: **Instance reweighting approaches the oracle. Left:** Gain over *pass@1* (right bar) and gap to *pass@8* (left bar); x-axis is symmetric in percentage points (smaller gap = closer to oracle). "Sanity check" uses the test set as the meta set to probe headroom. **Right:** Concept: domain reweighting assigns one weight per dataset, whereas *instance* reweighting (Instance Table/Net) weights each sample, yielding larger gains and a smaller oracle gap (gray arrow → oracle).

However, DreamPRM still shows a clear gap to the *oracle performance* (Fig. 1). We evaluate against a test-time-scaling upper bound—the oracle—given by pass@$N$: sample $N$ response chains per question and count success if any chain is correct. An ideal selector would always pick a correct chain whenever one exists (thus reaching oracle performance); conversely, a method that fails to improve over the pass@1 baseline is of little practical value. For the domain-reweighting method, the gap persists even under a *sanity check* that uses testing data to update domain weights. We attribute this shortfall to meta-level *under-parameterization*: a single weight per dataset is too coarse to capture substantial within-dataset heterogeneity, limiting exploitation of both meta and training sets and motivating increased meta capacity beyond domain-level weights.

Therefore, we propose **OraclePRM**, which extends domain-level reweighting to *instance-level* reweighting. Every training example receives an adaptive weight so that informative samples are amplified while trivial or noisy ones are suppressed. However, a naive implementation—instance weights optimized with bi-level updates—proves brittle in our experiment: training often diverges or oscillates. We trace this to a time-scale mismatch between upper-level and lower-level updates. Similar pathologies are well-documented in other bi-level systems (e.g., differentiable NAS) where outer-objective progress can coexist with degraded solutions, and stabilization typically requires explicit control of update schedules and capacity (Liu et al., 2019).

To make instance reweighting practical for multimodal PRM training, we introduce a stable instance-reweighting framework with two regimes (Fig.2): (i) **Instance Table** (small data): directly learn a per-sample weight for each training example, maximizing the utility of limited data. (ii) **Instance Net** (large data): learn a generalizable mapping from sample features to weights. Across both regimes we add cold-start initialization and bounded-range constraints (e.g., clip function and scaling sigmoid) to prevent runaway updates. In addition, we carry out a time-scale matching experiment to determine appropriate settings for critical hyperparameters (e.g., number of training samples and parameters). This recipe expands meta capacity without sacrificing stability.

Our contributions are summarized as follows:

- We propose *OraclePRM*, an instance-level reweighting framework for multimodal PRM training. It offers two complementary regimes—*Instance Table* (explicit per-sample weights for small/medium data) and *Instance Net* (lightweight, generalizable weight predictor for large-scale data)—together with a stable bi-level recipe (time-scale matching, cold-start initialization, bounded weights).

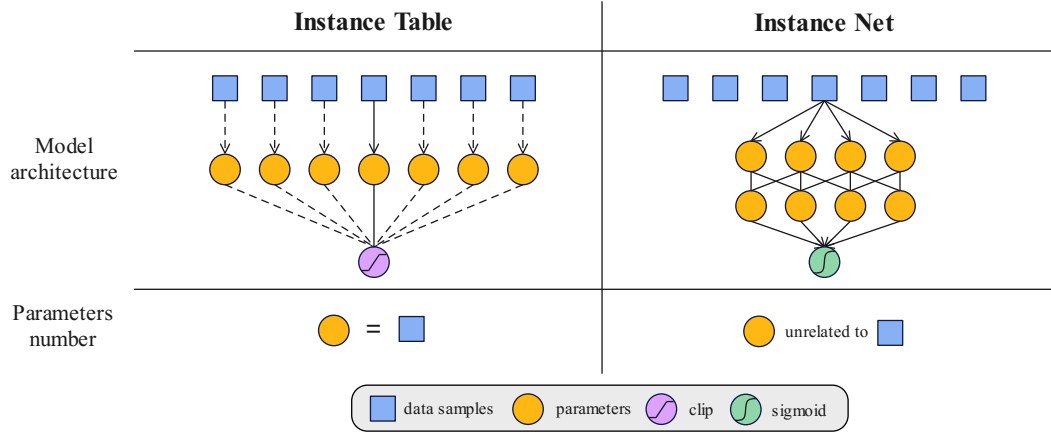

Figure 2: Comparison of two model designs for instance reweighting in OraclePRM. Instance Table assigns an explicit learnable weight to each training sample, offering strong per-instance flexibility but scaling with dataset size. Instance Net parameterizes instance weights via a lightweight MLP appended to the PRM, maintaining a fixed parameter size independent of dataset scale and providing better generalization.

- Integrated with test-time scaling, OraclePRM closes the sanity-check gap toward the oracle (*pass@8*), generalizes across multimodal reasoning benchmarks, and converges stably; it also sets a new SOTA (**84.6** on *MMMU* validation) and achieves **first-place** results when paired with a leading backbone (e.g., GPT-5-mini).

## 2 THE PROPOSED INSTANCE-REWEIGHTING METHOD

Training multimodal PRMs is difficult due to (1) data quality imbalance and (2) mismatch between training and inference. We propose OraclePRM, which learns instance weights via a bi-level framework adapted from DreamPRM (Cao et al., 2025) . The lower level updates PRM parameters with instance-reweighted training, while the upper level optimizes instance weights on a meta dataset.

### 2.1 BI-LEVEL OPTIMIZATION FOR INSTANCE REWEIGHTING

**Lower-level optimization.** We denote the Process Reward Model (PRM) as $\mathcal{V}_\phi$, parameterized by $\phi$, and assign each training instance $(x, y)$ a single learnable weight $\alpha_{(x,y)}$. Given a training set $\mathcal{D}_{tr} = \{(x, y)\}$, the training loss on one instance $(x, y)$ is defined as

$$\mathcal{L}_{tr}(\phi, \alpha; x, y) = \alpha_{(x,y)} \cdot \sum_{i=1}^{N} \mathcal{L}_{\text{CE}}\big(\mathcal{V}_\phi(x, \hat{y}_{\leq i}), \ p_i\big), \tag{1}$$

where $N$ is the number of reasoning steps in the trajectory $y$, $\hat{y}_{\leq i}$ denotes the prefix consisting of the first $i$ steps, and $p_i$ is the supervision signal for step $i$.

$$\phi^*(\alpha) = \arg\min_\phi \ \mathbb{E}_{x \sim \mathcal{D}_{tr}}\big[\mathcal{L}_{tr}(\phi, \alpha; x)\big]. \tag{2}$$

Here, only $\phi$ is updated while $\alpha$ remains fixed.

**Upper-level optimization.** We then refine $\alpha$ on the meta-dataset $\mathcal{D}_{meta}$ using a meta loss that emulates PRM inference. For each generated solution $\hat{y}$, we compute an aggregated trajectory score

$$s(x, \hat{y}; \phi^*(\alpha)) = \mathcal{A}\big(\mathcal{V}_{\phi^*(\alpha)}(x, \hat{y})\big), \tag{3}$$

where $\mathcal{A}(\cdot)$ aggregates step-level rewards into a single trajectory score. This score is then aligned with the ground-truth correctness label $r(\hat{y}, y) \in \{0, 1\}$ through

$$\mathcal{L}_{meta}(\phi^*(\alpha); \mathcal{D}_{meta}) = \sum_{(x,y) \in \mathcal{D}_{meta}} \mathcal{L}_{\text{MSE}}\Big(s(x, \hat{y}; \phi^*(\alpha)), \ r(\hat{y}, y)\Big), \tag{4}$$

Finally, the instance weights $\alpha$ are updated by gradient descent:

$$\alpha \leftarrow \alpha - \eta \nabla_\alpha \, \mathcal{L}_{meta}(\phi^*(\alpha); \mathcal{D}_{meta}), \tag{5}$$

with meta learning rate $\eta$.

**Discussion.** This bi-level scheme ensures that $\phi$ adapts to the reweighted training data, while $\alpha$ itself is refined to minimize validation error. As a result, OraclePRM adaptively emphasizes informative instances and suppresses noisy or redundant ones.

## 2.2 MODEL DESIGN FOR INSTANCE REWEIGHTING

**Instance Table.** A straightforward way to assign weights at the instance level is to maintain a lookup table, where each training example $x \in \mathcal{D}_{tr}$ is associated with a learnable scalar weight $\alpha_x$. Formally, the instance-level weighted loss can be written as

$$\mathcal{L}_{\text{IT}}(\phi, \alpha) = \sum_{(x,y) \in \mathcal{D}_{tr}} \alpha_x \cdot \ell\big(\mathcal{V}_\phi(x), y\big), \tag{6}$$

where $\ell(\cdot)$ is a step-wise cross-entropy loss. In this formulation, the number of parameters in $\alpha$ equals the number of training samples. This design allows maximum flexibility by fully exploiting the contribution of each example, often yielding strong performance even with relatively small datasets (see experiments). To avoid extreme values, we apply a clipping function

$$\alpha_x \leftarrow \text{clip}(\alpha_x, \alpha_{\min}, \alpha_{\max}), \tag{7}$$

which projects any out-of-range values back to the boundary $[\alpha_{\min}, \alpha_{\max}]$.

**Instance Net.** An alternative strategy is to parameterize instance weights using a lightweight neural network. Specifically, we introduce a small multi-layer perceptron (MLP) $f_\psi(\cdot)$ applied to the final hidden representation $h(x)$ of the PRM:

$$\alpha_x = c \cdot \sigma\big(f_\psi(h(x))\big), \tag{8}$$

where $\sigma(\cdot)$ is the sigmoid activation and $c > 1$ is a fixed scaling factor. This design expands the weight range from $(0, 1)$ to $(0, c)$, alleviating the constraint of overly small weights while still maintaining smooth boundedness. More generally, one can define

$$\alpha_x = a + (b - a) \cdot \sigma\big(f_\psi(h(x))\big), \tag{9}$$

which yields weights in $(a, b)$ for any specified interval $[a, b]$.

The corresponding weighted loss becomes

$$\mathcal{L}_{\text{IN}}(\phi, \psi) = \sum_{(x,y) \in \mathcal{D}_{tr}} \alpha_x \cdot \ell\big(\mathcal{V}_\phi(x), y\big). \tag{10}$$

Compared to the Instance Table, Instance Net has a fixed parameter budget independent of dataset size, while still flexibly scaling weights to match the desired range.

## 2.3 TECHNICAL DETAILS

**Generative reward model.** We employ a generative reward model to assign scores to individual reasoning steps. Specifically, we adapt the system prompt from VisualPRM (Wang et al., 2025) (See Appendix B.3), which instructs the model to output either + or − for each step in the response. The score is then computed as the softmax probability of the + token. A higher probability indicates greater model confidence in the correctness of the step, and thus corresponds to a higher reward score.

**Cold-start initialization.** Prior to bi-level optimization, we perform a cold-start fine-tuning stage. Specifically, we sample 20k examples from VisualPRM-400K (Wang et al., 2025) and conduct one epoch of supervised fine-tuning (SFT). The resulting checkpoint is then used as the initialization for bi-level optimization. This step ensures that the base model learns to follow the system prompt and reliably generate + and − tokens, which are essential for subsequent optimization.

**Multi-turn fine-tuning.** We cast process supervision as a multi-turn dialogue task to better exploit the generative capabilities of MLLMs. Given a multimodal input question $x$, the first turn includes the question and its initial reasoning step $\hat{y}_1$, while each subsequent turn introduces the next step in the reasoning trajectory. This formulation allows the model to incrementally process and evaluate reasoning steps in a conversational manner.

**Aggregation function loss.** Following DreamPRM (Cao et al., 2025), we adopt an aggregation loss for the meta-learning of the generative reward model. Specifically, we apply a mean aggregation function to average the step-level scores, and optimize the model using the mean squared error (MSE) between the aggregated score and the ground-truth binary label. To encourage the model to generate both + and −, we compute the score from the logit of + when the ground-truth label is positive, and from the logit of − when the ground-truth label is negative (denoted as "both loss" in ablation studies).

# 3 EXPERIMENTS

## 3.1 MAIN EVALUATIONS

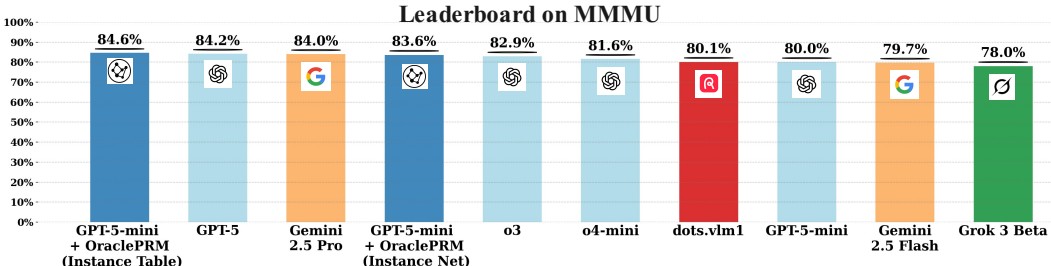

Figure 3: **Leaderboard on MMMU as of September 16, 2025.** Results are taken from the official MMMU leaderboard validation set (Yue et al., 2024), while the scores of OraclePRM (Instance Table and Instance Net) and the base GPT-5-mini are measured by us.

Table 1: Performance on MathVista, WeMath, MathVision, MMStar, OlympaidBench benchmarks, and Overall. Red numbers indicate absolute gains over the base model.

| Model | MathVista | WeMath | MATH−V | MMStar | OlympaidBench | Overall |
|---|---|---|---|---|---|---|
| InternVL3-8B | 71.4 | 57.7 | 30.8 | 67.0 | 34.2 | 52.2 |
| +OraclePRM-Instance Table | 72.2 | 62.4 | 33.1 | 69.4 | 38.4 | 55.1 |
| | (+0.8) | (+4.7) | (+2.3) | (+2.4) | (+4.2) | (+2.9) |
| +OraclePRM-Instance Net | 72.6 | 61.8 | 33.7 | 69.9 | 39.2 | 55.4 |
| | (+1.2) | (+4.1) | (+2.9) | (+2.9) | (+5.0) | (+3.2) |
| Qwen2.5-VL-7B | 68.2 | 64.4 | 30.5 | 63.8 | 33.6 | 52.1 |
| +OraclePRM-Instance Table | 70.3 | 64.6 | 33.7 | 65.8 | 37.6 | 54.4 |
| | (+2.1) | (+0.2) | (+3.2) | (+2.0) | (+4.0) | (+2.3) |
| +OraclePRM-Instance Net | 69.8 | 68.7 | 32.9 | 65.8 | 40.6 | 55.6 |
| | (+1.6) | (+4.3) | (+2.4) | (+2.0) | (+7.0) | (+3.5) |
| InternVL-2.5-8B-MPO | 65.4 | 51.7 | 20.4 | 58.9 | 10.0 | 41.3 |
| +OraclePRM-Instance Table | 68.3 | 57.9 | 22.1 | 61.2 | 16.2 | 45.1 |
| | (+2.9) | (+6.2) | (+1.7) | (+2.3) | (+6.2) | (+3.8) |
| +OraclePRM-Instance Net | 69.5 | 56.2 | 22.3 | 62.0 | 17.7 | 45.5 |
| | (+4.1) | (+4.5) | (+1.9) | (+3.1) | (+7.7) | (+4.2) |

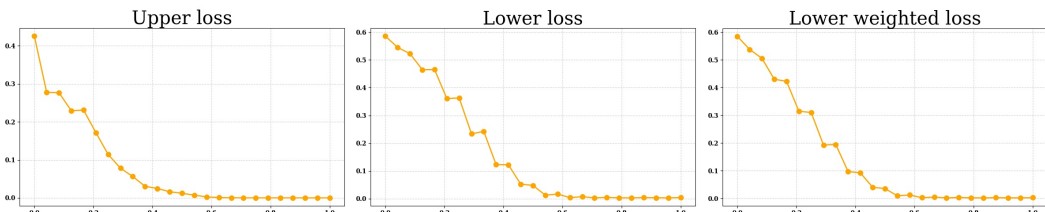

Figure 4: **Training loss of instance table.** The figure shows the convergence of the training loss, indicating stable optimization and steady improvement over time.

**Weight Distribution Evolution**

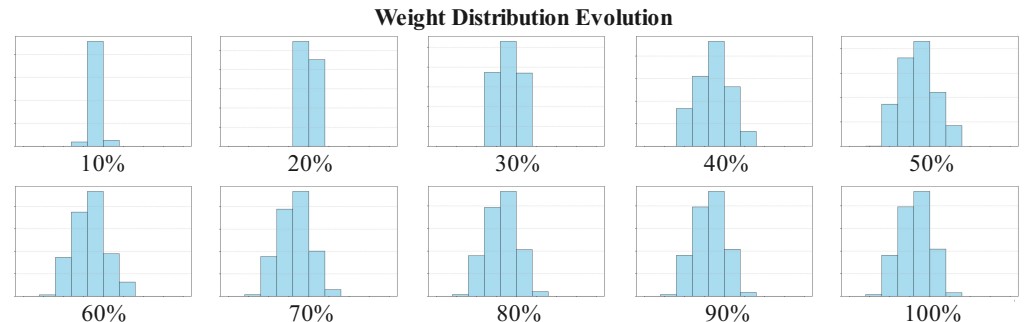

Figure 5: **Learned weight distributions of instance table.** The figure shows the distribution of learned domain weights at different training stages. The x-axis ranges from 0 to 2, with each bin corresponding to an interval of 0.2.

Table 2: Comparison of different baseline methods on MMMU validation set. $\triangle$ indicates absolute gains over the base model.

| Category | Method | Accuracy | $\triangle$ |
|---|---|---|---|
| Base Model | GPT-5-mini w/ thinking | 80.0 | – |
|  | + Self-Consistency (Wang et al., 2023) | 81.4 | +1.4 |
| Training with Data Selection | No selection | 79.1 | -0.9 |
|  | s1 selection (Muennighoff et al., 2025) | 80.2 | +0.2 |
| Other PRMs | DreamPRM (Cao et al., 2025) | 81.2 | +1.2 |
|  | VisualPRM (Wang et al., 2025) | 80.5 | +0.5 |
| OraclePRM | Instance Table | **84.6** | **+4.6** |
|  | Instance Net | 83.6 | +3.6 |

**Evaluation protocol.** Our main evaluation tests both OraclePRM variants (**Instance Table** and **Instance Net**) and is organized into three parts: (i) leaderboard performance, (ii) benchmark evaluation, and (iii) baseline comparison. For all experiments, we use Best-of-$N$ testing ($N$=8): we sample eight responses per question, let the PRM select the best candidate, and report improvement over the pass@1 baseline. In the main evaluation, Instance Table is trained with 10k lower-level training samples, whereas Instance Net is trained with 100k. Implementation details are provided in Appendix B.

**Leaderboard performance.** We evaluate the performance of our methods on MMMU validation set (Yue et al., 2024). Fig. 3 summarizes the results on the MMMU leaderboard. Using GPT-5-mini with thinking as our base model (80.0 accuracy), OraclePRM significantly improves performance and achieve SOTA results. Both Instance Table and Instance Net variants achieve substantial gains of +4.6 and +3.6, respectively, demonstrating the effectiveness of our approach even on the most advanced

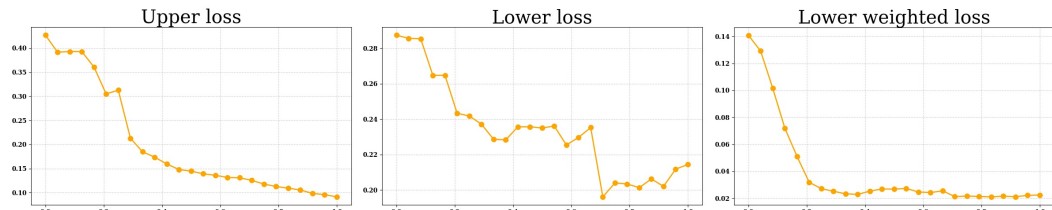

Figure 6: **Training loss curve of instance net.** The upper panel shows the convergence of the training loss, indicating stable optimization and steady improvement over time.

**Weight Distribution Evolution**

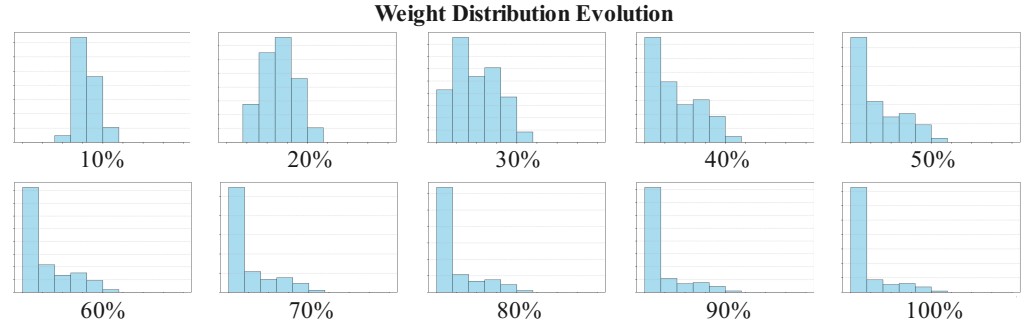

Figure 7: **Learned weight distributions of instance net.** The figure shows the distribution of learned domain weights at different training stages. The x-axis ranges from 0 to 2, with each bin corresponding to an interval of 0.2.

model and in the most competitive comparison. More details of the leaderboard performance are provided in Appendix C.

**Benchmark evaluation.** Table 1 presents results on five multimodal reasoning benchmarks across three different backbone models. First, our method demonstrates strong generalization: regardless of the underlying model (InternVL3-8B (Zhu et al., 2025), Qwen2.5-VL-7B (Wang et al., 2024d), or InternVL-2.5-8B-MPO (Wang et al., 2024e)) and the benchmark (WEMATH (Qiao et al., 2024), MATHVISTA (Lu et al., 2024), MATHVISION (Wang et al., 2024b) OLYMPAIDBENCH (He et al., 2024) and MMSTAR (Chen et al., 2024a)), OraclePRM consistently brings stable improvements over the base models. This confirms that the proposed data reweighting strategy is effective across both architectures and tasks. Second, we observe that OraclePRM–Instance Net performs slightly better than OraclePRM–Instance Table. Interestingly, this is the opposite of the leaderboard trend (cf. Fig. 3), where Instance Table had the advantage. This reversal suggests that training with larger and more diverse data enhances the generalization ability of Instance Net, allowing it to adapt better to different benchmarks. Together, these results highlight both the robustness and transferability of our approach. More details of the benchmark evaluation are provided in Appendix D.

**Baseline comparasion.** Table 2 compares OraclePRM with several representative baselines. First, Self-Consistency (Wang et al., 2023) is the most widely used test-time scaling method. Our approach consistently surpasses self-consistency, demonstrating that OraclePRM provides more effective performance gains beyond what can be achieved by sampling-based inference alone. Second, since data reweighting can be viewed as a dynamic form of data selection, we also compare against static selection strategies. Without any selection, performance even drops below the base model, highlighting that noisy lower-level data indeed harms training. In contrast, appropriate selection or reweighting leads to clear improvements, confirming the importance of handling data quality. Third, we further compare with other recently proposed PRM methods, such as DreamPRM (Cao et al., 2025) and VisualPRM (Wang et al., 2025). Their improvements are relatively minor, whereas OraclePRM yields much larger gains. This indicates that our upper-level instance-aware reweighting provides a more powerful correction mechanism, enabling more substantial performance improvements. More details of the baseline comparison are provided in Appendix E.

Table 3: Accuracy across different data sample sizes. Best results are highlighted in bold.

| Category | Samples | Accuracy | Sanity check |
|---|---|---|---|
| | 1000 | 82.4 | **88.7** |
| Instance Table | 10 000 | **83.8** | **88.7** |
| | 100 000 | 81.2 | 82.5 |
| | 1000 | 80.9 | **89.1** |
| Instance Net | 10 000 | 82.0 | 87.1 |
| | 100 000 | **83.6** | 85.0 |

Table 4: Accuracy with different parameter scales across Domain Table, Instance Table, and Instance Net variants (100k lower-level training samples).

| Category | Parameters | Accuracy | Sanity check |
|---|---|---|---|
| Domain Table | 20 | 80.9 | 84.8 |
| Instance Table | 100 000 | 81.2 | 82.5 |
| | 41 (hidden dim = 10) | **83.6** | 85.0 |
| Instance Net | 121 (hidden dim = 30) | 83.5 | 86.8 |
| | 201 (hidden dim = 50) | 83.1 | **87.5** |

## 3.2 MORE DISCUSSIONS

**Experiment design.** Beyond accuracy, this subsection introduces three diagnostics for evaluating OraclePRM: (i) **training loss curves** to assess convergence and the synchronization of upper-lower updates; (ii) **weight–distribution analyses** to check whether weights converge and meaningfully differentiate samples (rather than collapsing or not changing); and (iii) a **sanity check** that uses test-distribution meta data to estimate the attainable upper bound. We further use these diagnostics to conduct *time-scale matching* and *ablation studies*, validating key hyperparameters and modules.

**Training loss curve.** We report the training loss curves for both the Instance Table (Fig. 4) and the Instance Net (Fig. 6). In both cases, the methods eventually converge, with the upper and lower losses converging simultaneously, demonstrating the effectiveness of our training procedure. For Instance Table, the average weighted loss per epoch closely follows the lower loss. This indicates that some samples receive weights greater than 1 while others receive weights less than 1, showing that the instance table effectively differentiates sample importance. This behavior is further confirmed by the distribution of learned weights. In contrast, Instance Net exhibits a larger discrepancy between the weighted loss and the lower loss. This is because the Instance Net dynamically adjusts weights throughout different stages of training, with the weight allocation varying in accordance with the upper loss. This highlights a key distinction between the two approaches: Instance Net behaves like a "loss scheduler," adapting weights based on the training process, while Instance Table assigns weights primarily according to data quality. Both strategies contribute to more efficient training.

**Weight distributions.** We report the weight distributions of the Instance Table (Fig. 5) and Instance Net (Fig. 7) across different stages of training. For the Instance Table, the weights are initially concentrated around 1, but gradually spread across the range of 0–2. After approximately 60% of training, the distribution stabilizes, indicating convergence. This observation is consistent with the loss curve, which also remains stable beyond 60% of training. In contrast, Instance Net assigns larger weights during the early training stage, but these weights eventually diminish toward 0 as the outer loss decreases, suggesting that training has already become sufficient. This behavior supports our hypothesis that the instance net functions like a "loss scheduler," dynamically adjusting weights according to the training process.

Table 5: Ablation study on different training strategies.

| Ablation | Instance Table | | Instance Net | |
|---|---|---|---|---|
| | Accuracy | Sanity check | Accuracy | Sanity check |
| w/o cold start | 82.2 (-1.6) | 88.0 (-0.7) | 81.9 (-1.7) | 86.0 (+1.0) |
| w/o both loss | 82.6 (-1.2) | 85.2 (-3.5) | 83.3 (-0.3) | 86.3 (+1.3) |

**Sanity check.**   We additionally conduct a *sanity check*, where the meta set used for reweighting is directly taken from the test set (Appendix Fig. 9). In this setting, the instance weights are fine-tuned using meta feedback from the test distribution. Since the meta set only serves to control the training of instance weights—while the PRM itself is still trained solely on the training set—this experiment does not involve direct information leakage into the PRM. The results show that OraclePRM nearly matches the oracle upper bound, confirming that the performance gap mainly stems from distribution shift between the meta set and the test set. These findings highlight OraclePRM's potential to approach oracle-level performance when the meta distribution aligns with the target evaluation. More details of the sanity check can be found in Appendix F.

**Time-scale matching.**   To maintain balanced update rates between the upper and lower levels, we systematically vary critical hyperparameters (e.g., training-sample and number of parameters) and select the effective configuration. In this evaluation, we split the MMMU validation set in half: one half is used as the meta set and the other half as the test set. We also report the sanity check results on this held-out portion. This setting is chosen because the smaller meta set allows for faster convergence, enabling us to run more experiments and tune hyperparameters more effectively before conducting official training with a larger set. Each experiment in this setup is run for 50,000 iterations.

DATA SAMPLE NUMBER.   Table 3 shows that Instance Table achieves its best performance with 10k samples, while using 100k leads to degradation, suggesting time-scale mismatch. In contrast, Instance Net benefits from larger-scale training, reaching its best accuracy at 100k samples (83.6), consistent with its dynamic weighting design that adapts well to more data. Therefore, we adopt 10k samples for Instance Table and 100k for Instance Net. More details are provided in Appendix G.

PARAMETER NUMBER.   To examine the trade-off between capacity and performance, we analyze different parameterizations of Domain Table, Instance Table, and Instance Net on 100k lower-level samples (Table 4). Domain Table uses only 20 parameters, while Instance Table scales up to 100k. For Instance Net, we vary hidden dimensions from 10 to 50 and observe that accuracy remains stable at 83.6 when increasing from 10 to 30, indicating that extra capacity does not directly yield better performance. These results suggest that lightweight reweighting modules already capture most of the useful supervision, and further gains may require more data or stronger regularization. More details are provided in Appendix H.

**Ablation studies.**   To assess the contribution of each component in our framework, we conducted an ablation study (Table 5). Removing the cold-start initialization ("no cold start") tests whether the model can still learn effective weights without carefully designed initialization. Excluding the auxiliary joint objective ("no both loss") isolates the impact of multi-level supervision. More details of ablation studies are provided in Appendix I.

## 4 CONCLUSION

In this paper, we presented OraclePRM, an instance-reweighted multimodal PRM framework that extends domain-level reweighting in DreamPRM to the level of individual training examples. Through bi-level optimization, OraclePRM dynamically learns instance weights that emphasize informative samples while down-weighting noisy or trivial ones. We further explored two complementary implementations—Instance Table and Instance Net—that trade off per-sample expressiveness and scalability. Extensive experiments on the MMMU benchmark demonstrated that OraclePRM significantly improves GPT-5-mini, achieving state-of-the-art performance.

ETHICS STATEMENT

This work complies with the ICLR Code of Ethics. Our study involves only publicly available multimodal reasoning datasets (e.g., MathVista, WeMath, MathVision, MMStar, OlympiadBench, MMMU, VisualPRM-400K) and models; no human subjects, personal information, or sensitive attributes are used, so IRB approval was not required. All datasets were used under their licenses, and we document sources and licensing in the references and appendix. To mitigate potential data leakage, we adopt strict filtering procedures—including duplicate, partial-overlap, and semantic similarity filters—as described in Appendix B. Our methods target research and education, and should not be misused (e.g., to automate academic assessments). We declare no conflicts of interest or external sponsorship.

REPRODUCIBILITY STATEMENT

We have made every effort to ensure the reproducibility of our results. The training details, hyper-parameters, and model architectures are fully described in Section 2 and Appendix B. We provide algorithm descriptions in Section 2. All datasets used are publicly available, and preprocessing steps are detailed in Appendix B. We also provide the source code and scripts for data processing and experiments in the supplementary material.

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

# APPENDIX

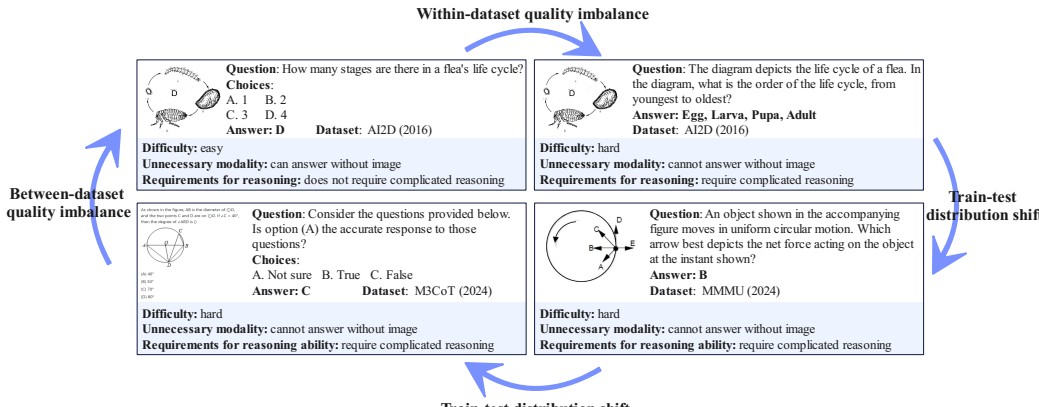

Figure 8: **Sources of data mismatch in multimodal reasoning.** The examples illustrate three challenges: (i) *within-dataset quality imbalance*—the same dataset (AI2D (Kembhavi et al., 2016)) contains easy items solvable without images versus hard items requiring visual reasoning; (ii) *between-dataset quality imbalance*—different datasets (e.g., M3CoT (Chen et al., 2024b) vs. AI2D) vary in difficulty and modality necessity; and (iii) *train–test distribution shift*—training corpora are often skewed toward easier, low-vision items while test benchmarks demand complex, image-dependent reasoning. Each card reports difficulty, modality necessity, and reasoning requirements. These phenomena motivate reweighting (domain- and instance-level) for robust PRM training.

# A    RELATED WORKS

**Multimodal Reasoning**    Recent works have highlighted the importance of Chain-of-Thought (CoT) reasoning (Wei et al., 2022; Kojima et al., 2022; Zhang et al., 2023) in large language models (LLMs), which encourages a step-by-step approach and significantly boosts question-answering accuracy. However, directly transferring CoT prompting to multimodal LLMs (MLLMs) remains challenging due to hallucinations and inconsistencies in generated reasoning chains (Wang et al., 2024e; Zheng et al., 2024; Jiang et al., 2025). To address these limitations, post-training methods have been proposed. InternVL-MPO (Wang et al., 2024e) introduces mixed preference optimization that jointly optimizes preference ranking, response quality, and generation objectives. Llava-CoT (Xu et al., 2024) constructs structured reasoning datasets to explicitly align MLLMs with systematic step-by-step thinking. At inference time, RLAIF-V (Yu et al., 2024a) leverages self-feedback guidance and length normalization for scaling reasoning quality, while AR-MCTS (Dong et al., 2024) combines Monte Carlo Tree Search (MCTS) with Retrieval-Augmented Generation (RAG) to guide exploration in the reasoning space.

**Process Reward Models**    Process Reward Models (PRMs) (Lightman et al., 2024; Li et al., 2023; Ma et al., 2023; Wang et al., 2024a) provide step-level verification of reasoning trajectories, offering finer granularity compared to Outcome Reward Models (ORMs) (Cobbe et al., 2021; Shao et al., 2024). A key difficulty lies in collecting reliable supervision signals for each reasoning step, which often relies on costly manual annotation (Lightman et al., 2024). To reduce annotation cost, Math-Shepherd (Wang et al., 2024c) leverages Monte Carlo estimation to generate both hard and soft process labels, while OmegaPRM (Luo et al., 2024) integrates MCTS for fine-grained trajectory exploration. MiPS (Wang et al., 2024f) further refines Monte Carlo–based aggregation of PRM signals. More recently, VisualPRM (Wang et al., 2025) introduced process-level supervision and generative reward model to multimodal reasoning, and open-sourced VisualPRM 400K for multimodal PRM tarining. DreamPRM (Cao et al., 2025) proposed a data-reweighting framework to automatically emphasize high-quality reasoning samples while suppressing noisy ones, and demonstrated that such reweighting significantly improves test-time scaling performance across diverse multimodal benchmarks. OraclePRM extends these advances by introducing instance-level reweighting strategies that further enhance supervision fidelity and generalization.

**Data Reweighting**   Data reweighting techniques are widely studied to modulate the influence of heterogeneous training data, ensuring robust generalization across domains. In large-scale pretraining, DoReMi (Xie et al., 2023) introduces a proxy-based optimization scheme that assigns domain weights via distributionally robust optimization. DOGE (Fan et al., 2024a) formulates a first-order bi-level optimization approach, where mixture weights are updated through gradient alignment between source and target domains. Data Mixing Laws (Ye et al., 2025) complement these methods by deriving analytical scaling laws to predict performance under different mixture strategies. DreamPRM adapted data reweighting to process supervision, showing its effectiveness in balancing quality across reasoning trajectories. In this work, OraclePRM generalizes the idea further by moving from domain-level to instance-level reweighting, introducing scalable mechanisms such as instance tables and instance nets to dynamically assign weights to each training example.

## B   IMPLEMENTATION DETAILS

### B.1   MODEL.

We adopt InternVL3-1B (Zhu et al., 2025) as the base model for training the PRM. This state-of-the-art, small-scale multimodal model is pretrained on general vision–language understanding tasks, and we fine-tune it to obtain the final checkpoint. For inference, we use GPT-5-mini (OpenAI et al., 2024) as the underlying MLLM. GPT-5-mini is a lightweight variant of the state-of-the-art reasoning model GPT-5, offering a favorable balance between cost efficiency and competitive performance.

### B.2   TRAINING AND META DATASETS.

For training, we construct two datasets. Specifically, we sample 12k examples from VisualPRM-400k (Wang et al., 2025) to train the Instance Table variant of instance reweighting, and 100k examples from the same source to train the Instance Net variant. We also conduct a rule-based check to ensure there is no overlap between training set and test set.

For the meta set, we adopt MMMU-Pro (Yue et al., 2024) (standard 4-option split), while excluding its validation split to avoid overlap. We further use GPT-5-mini to generate four candidate responses for each question, forming the meta-evaluation set used for weight updates. There are about 1.2k data points in meta set.

To maintain balance between positive and negative supervision, we filter both the training and meta datasets to ensure an approximately equal number of positive and negative samples.

### B.3   SYSTEM PROMPT FOR GENERATIVE REWARD MODEL.

The Generative Reward Model leverages a system prompt adapted from VisualPRM (Wang et al., 2025).

```
You are an advanced AI assistant, designed to serve as a process
supervision model.  In this task, I will provide a problem
statement followed by the first step of the solution process.  For
each subsequent turn, I will give you a new step in the solution.
Your role is to assess whether the solution process is correct up
to the current step.- In the **first round**, I will input the
problem and the first step of the solution process.- In **each
subsequent round**, I will provide the next step in the solution.
For each step, you should:- Respond with **+** if you believe the
solution process is correct up to this step.- Respond with **-**
if you detect any issues or errors in the process up to this step.
Please note:- Only respond with **+** or **-**.  Do not provide
any additional explanations, comments, or justifications.  Your
task is to verify the accuracy and correctness of each step in the
given solution process.
```

Table 6: Performance on the MMMU validation set (900 examples). Starred numbers (*) are reported by the corresponding authors.

| Model | MMMU-Pro | MMMU (Val) |
|---|---|---|
| Human Expert (High) | 85.4 | 88.6 |
| GPT-5 w/ thinking | 78.4* | 84.2* |
| Gemini 2.5 Pro Deep-Think | - | 84.0* |
| o3 | 76.4* | 82.9* |
| Human Expert (Medium) | 80.8 | 82.6 |
| o4-mini | - | 81.6* |
| dots.v1m1 (37B) | 70.1* | 80.1* |
| Gemini 2.5 Flash 05-20 | - | 79.7* |
| Gemini 2.5 Pro 05-06 | - | 79.6* |
| o1 | - | 78.2* |
| Grok 3 Beta | - | 78.0* |
| Seed 1.5-VL Thinking (20B) | 67.6* | 77.9* |
| Intern-S1 (22B) | - | 77.7* |
| Claude Opus 4.1 | - | 77.1* |
| Claude Opus 4 | - | 76.5* |
| Human Expert (Low) | 73.0 | 76.2 |
| Llama 4 Behemoth (288B) | - | 76.1* |
| Skywork-R1V3-38B (38B) | 55.4* | 76.0* |
| GLM-4.5V w/ Thinking (12B) | 65.2* | 75.4* |
| Claude 3.7 Sonnet | - | 75.0* |
| Seed 1.6-Thinking (20B) | 66.4* | 74.8* |
| GPT-5 w/o thinking | 62.7* | 74.4* |
| GPT-4.5 | - | 74.4* |

## B.4 HYPERPARAMETER SETTINGS.

For the lower-level optimization, we perform one inner gradient step per outer update (*unroll steps* = 1), using the AdamW optimizer (Loshchilov & Hutter, 2019) with a learning rate of $5 \times 10^{-5}$ and weight decay of $10^{-2}$.

For the upper-level optimization, we also adopt AdamW. In the Instance Table setting, we use a learning rate of $5 \times 10^{-3}$ with weight decay $10^{-3}$; in the Instance Net setting, we use a learning rate of $5 \times 10^{-4}$ with weight decay $10^{-3}$, and set the hidden dimension of the network to 10. The weight range of Instance Table and Instance Net is set to $[0, 2]$.

Both levels employ a cosine learning rate schedule with linear warm-up, where the warm-up phase corresponds to 5% of the total training steps. Overall, OraclePRM is fine-tuned for 150,000 iterations. The framework is implemented using Betty (Choe et al., 2023), and full training requires approximately 72 hours on a single 80GB NVIDIA A100 GPU.

## B.5 SUBSAMPLING WITH FILTERING.

To prevent training set overlap with test sets, we apply three filters:

- `is_duplicate_match`: removes samples that are identical to any test instance.
- `is_partial_match`: removes samples with partial overlap on key text segments.
- `is_similarity_match`: uses `difflib.SequenceMatcher` with a threshold of 0.7 to remove semantically similar cases.

These filters effectively ensure that no data leakage occurs between training and test sets.

## C LEADERBOARD PERFORMANCE DETAILS

MMMU (Yue et al., 2024) is a recently introduced benchmark designed to evaluate multimodal models on large-scale, multi-disciplinary tasks that require college-level subject knowledge and

Table 7: Results (%) on math-related benchmarks for InternVL-3-8B, QwenVL-2.5-7B, and InternVL-2.5-8B-MPO. Each row corresponds to an independent run.

| Model / Run | MathVista | WeMath | MathVision | MMStar | OlympiadBench |
|---|---|---|---|---|---|
| *InternVL-3-8B* | | | | | |
| Run-1 | 71.4 | 57.7 | 30.8 | 67.0 | 34.2 |
| Run-2 | 70.5 | 57.0 | 30.5 | 67.7 | 33.5 |
| Run-3 | 70.9 | 57.4 | 30.5 | 67.0 | 34.7 |
| Run-4 | 70.7 | 58.0 | 30.7 | 67.0 | 36.6 |
| Run-5 | 71.2 | 57.1 | 30.0 | 67.6 | 36.6 |
| Run-6 | 70.9 | 57.7 | 30.9 | 66.5 | 32.8 |
| Run-7 | 71.9 | 57.2 | 30.5 | 67.2 | 35.8 |
| Run-8 | 70.6 | 59.0 | 30.6 | 66.3 | 36.8 |
| *QwenVL-2.5-7B* | | | | | |
| Run-1 | 68.2 | 64.4 | 30.5 | 63.8 | 33.6 |
| Run-2 | 68.5 | 64.5 | 31.2 | 62.9 | 35.2 |
| Run-3 | 68.7 | 65.1 | 29.8 | 64.1 | 36.2 |
| Run-4 | 68.9 | 64.5 | 29.8 | 64.2 | 34.4 |
| Run-5 | 68.5 | 63.5 | 30.6 | 65.2 | 35.1 |
| Run-6 | 69.1 | 65.0 | 30.3 | 63.7 | 32.7 |
| Run-7 | 69.7 | 64.8 | 30.6 | 63.6 | 33.6 |
| Run-8 | 66.8 | 64.7 | 29.9 | 63.7 | 36.6 |
| *InternVL-2.5-8B-MPO* | | | | | |
| Run-1 | 66.0 | 52.7 | 20.2 | 58.7 | 10.0 |
| Run-2 | 65.4 | 51.7 | 20.4 | 58.9 | 12.7 |
| Run-3 | 66.1 | 53.7 | 20.6 | 59.3 | 10.0 |
| Run-4 | 65.8 | 53.1 | 20.4 | 59.4 | 8.7 |
| Run-5 | 65.7 | 51.3 | 21.2 | 58.7 | 7.3 |
| Run-6 | 67.0 | 51.0 | 19.7 | 57.2 | 10.7 |
| Run-7 | 65.5 | 52.4 | 20.3 | 58.2 | 10.7 |
| Run-8 | 67.7 | 50.2 | 20.5 | 58.7 | 11.3 |

deliberate reasoning. It contains carefully curated multimodal questions sourced from college exams, quizzes, and textbooks, spanning six core disciplines: Art & Design, Business, Science, Health & Medicine, Humanities & Social Science, and Tech & Engineering. In total, the benchmark covers 30 subjects across 183 subfields, with questions paired with 30 diverse image types, including charts, diagrams, maps, tables, music scores, and chemical structures.

Table 6 summarizes the latest performance of state-of-the-art models on the MMMU validation set (900 examples). Human experts achieve an upper bound of 88.6%, while high-performing closed-source systems such as GPT-5 with thinking and Gemini 2.5 Pro Deep-Think reach above 84%. Among compact open-weight models, o3 and o4-mini also perform competitively, with scores above 81%. Notably, DreamPRM-1.5, built upon GPT-5-mini, attains 84.6%, which is close to the frontier models, indicating the effectiveness of data reweighting in enhancing reasoning capabilities. Overall, the validation results provide a reliable proxy for final test performance and highlight the strong impact of process supervision combined with reweighting methods.

# D BENCHMARK EVALUATION DETAILS

## D.1 BENCHMARKS USED IN THIS PAPER

**MathVista.** MathVista is a consolidated benchmark for mathematical reasoning in visual contexts. It contains 6,141 problems drawn from 28 existing multimodal datasets and three newly created subsets (IQTest, FunctionQA, PaperQA), covering chart/figure understanding, geometry, textbook QA, and compositional visual–math reasoning. We apply test-mini part in our evaluation, which contains 1000 questions.

**WeMath.** We−Math is a visual mathematics benchmark with ∼6.5K problems, organized by 67 hierarchical knowledge concepts and five levels of knowledge granularity. It decomposes composite

Table 8: MMMU accuracy (%) under *Self-Consistency* for different Best-of-$N$ (BoN).

| Model | BoN=1 | BoN=8 | BoN=16 | BoN=32 | BoN=64 | BoN=128 |
|---|---|---|---|---|---|---|
| InternVL2.5-8B | 56.2 | 58.0 | 58.6 | 60.4 | 59.7 | 60.6 |
| MiniCPM-V2.6 | 49.8 | 51.8 | 51.7 | 52.2 | 51.7 | 53.2 |

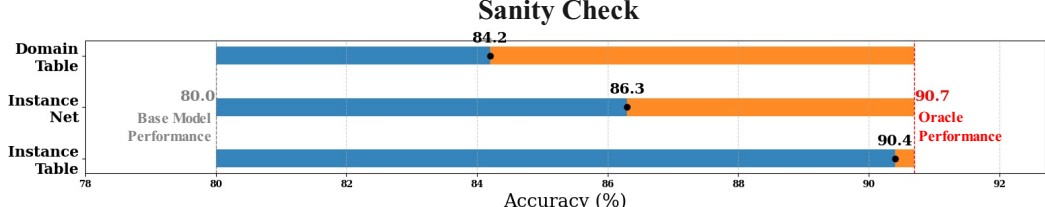

Figure 9: **Sanity check.** The figure shows sanity check performance of domain table method from DreamPRM, and our proposed instance net and instance table.

questions into sub-problems and reports four diagnosis-oriented metrics (Insufficient Knowledge, Inadequate Generalization, Complete Mastery, and Rote Memorization) to analyze model errors beyond end-to-end accuracy. We use losse accuracy in our evaluation.

**MathVision (MATH−V).** MathVision comprises 3,040 competition-style visual math problems spanning 16 disciplines and five difficulty levels. It emphasizes diverse, rigorously curated items to probe gaps between LMMs and human solvers on multimodal mathematical reasoning.

**MMStar.** MMStar is a vision-indispensable multimodal benchmark with 1,500 human-selected samples, designed to evaluate six core capabilities along 18 fine-grained axes using balanced and carefully purified items.

**OlympiadBench.** For OlympiadBench, the full dataset consists of 18 subsets, including multilingual and purely textual variants. To ensure a fair and focused evaluation aligned with our multimodal setup, we report results on the *OE_MM_maths_en_COMP* subset, which contains English multimodal math problems with image and text components.

## D.2 DESCRIPTION OF THE RESULT TABLES

Tables 7 (and related tables in the appendix) report per−run accuracies (in %) for *InternVL−3*, *QwenVL−2.5*, and *InternVL−2.5−MPO* on four multimodal math benchmarks (MathVista, We−Math, MathVision, MMStar). For each model we list eight independent runs under the same inference setting to expose run-to-run variability.

InternVL−2.5−MPO is trained with *Direct Preference Optimization (DPO)*, a preference-based objective that aligns model outputs with human preference data by directly optimizing the log-likelihood ratio between preferred and dispreferred responses. This setting avoids explicit reinforcement learning and enables stable alignment through supervised fine-tuning on preference pairs.

By contrast, InternVL−3 employs a reinforcement learning framework, where reward models guide multi-step optimization with explicit exploration. Such reinforcement learning provides stronger long-horizon credit assignment but usually requires more complex optimization (e.g., PPO-style updates or actor−critic loops). Empirically, the RL-based InternVL−3 achieves higher peak performance on reasoning-heavy tasks, while the DPO-based InternVL−2.5−MPO offers a simpler and more efficient alignment procedure that still yields competitive results.

## E   BASELINE COMPARISON DETAILS

**Self-Consistency.**   Self-Consistency (Wang et al., 2023) is an inference-time technique originally proposed for chain-of-thought prompting. Instead of relying on a single sampled reasoning path, the

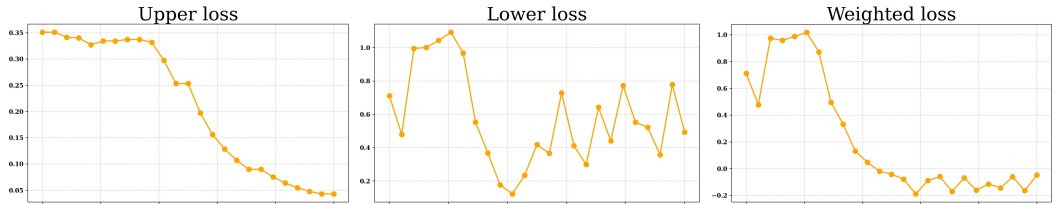

Figure 10: **Training loss curve of Instance Table — 1000 samples.** The upper panel shows the training loss.

method draws multiple reasoning trajectories, each potentially exploring different logical steps, and aggregates them via majority voting on the final answers. This approach mitigates the variance of individual generations, corrects for spurious or inconsistent reasoning paths, and better captures the distribution of plausible solutions. In multimodal reasoning tasks such as MMMU, Self-Consistency serves as a strong baseline for test-time scaling, often bridging a significant portion of the gap between base models and oracle performance.

Table 8 reports the performance of *InternVL2.5-8B* and *MiniCPM-V2.6* (Yao et al., 2024) on the MMMU benchmark under the *Self-Consistency* decoding strategy (Wang et al., 2025). We evaluate across different Best-of-$N$ (BoN) settings, where the model samples $N$ reasoning chains per problem and selects the most frequent final answer. As shown in the table, InternVL2.5-8B improves steadily from 56.2% at BoN=1 to 60.6% at BoN=128, while MiniCPM-V2.6 increases from 49.8% to 53.2% under the same setting. This demonstrates that larger candidate pools combined with majority voting can substantially improve robustness and accuracy in multimodal mathematical reasoning.

**s1 Data Selection Method** The s1 data selection was curated through a three-stage selection procedure guided by the principles of *quality*, *difficulty*, and *diversity* (Muennighoff et al., 2025).

**Quality filtering.** The selction began by collecting problems with reasoning traces. Datasets with formatting errors, low clarity, or noisy annotations were discarded.

**Difficulty filtering.** To ensure non-triviality, two models (Qwen2.5-7B-Instruct and Qwen2.5-32B-Instruct) were evaluated on each question. Items solved correctly by either model were discarded as too easy. In addition, reasoning trace length was used as a proxy for difficulty, under the assumption that harder problems induce longer traces.

**Diversity filtering.** To encourage broad coverage, each problem was categorized into domains based on the Mathematics Subject Classification (MSC) taxonomy (geometry, algebra, physics, economics, etc.). The final 1,000 samples were drawn by repeatedly sampling domains uniformly, and within each domain, selecting problems biased toward longer reasoning traces to maintain difficulty.

## F    SANITY CHECK DETAILS

We report the sanity check performance on the Domain Table ($\tilde{2}0$ parameters), Instance Net ($\tilde{4}1$ parameters), and Instance Table ($\tilde{1}0,000$ parameters) (Fig. 9). The results show that models with more trainable parameters achieve higher sanity check scores. This indicates that the upper-bound performance of data-reweighting training is closely tied to the capacity of higher-level trainable parameters. Moreover, sanity checks provide an important metric in our subsequent discussions.

## G    MORE DISCUSSIONS ON DATA SAMPLE NUMBER

**Instance Table — 1000 samples.** Fig. 10 and 11 report the training dynamics of the *Instance Table* variant with 1,000 training samples. Despite an initial decrease of the upper-level objective, the overall trajectory does not converge: the lower-level loss exhibits persistent oscillations and the weighted loss fails to stabilize, indicating a lack of joint progress across levels. This pattern is a hallmark of a *time-scale mismatch* in bi-level optimization: the inner (weight) dynamics evolve much faster than the outer (PRM) parameters, so the two objectives improve on different schedules and do

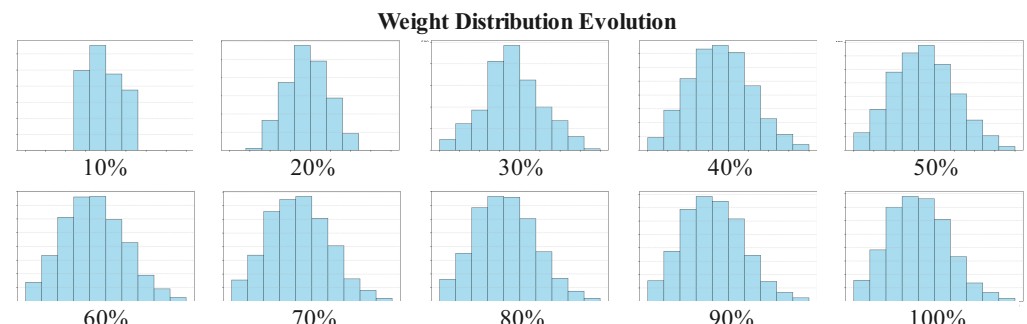

Figure 11: **Learned weight distributions of Instance Table — 1000 samples.** The figure shows the distribution of learned domain weights at different training stages. The x-axis ranges from 0 to 2, with each bin corresponding to an interval of 0.2.

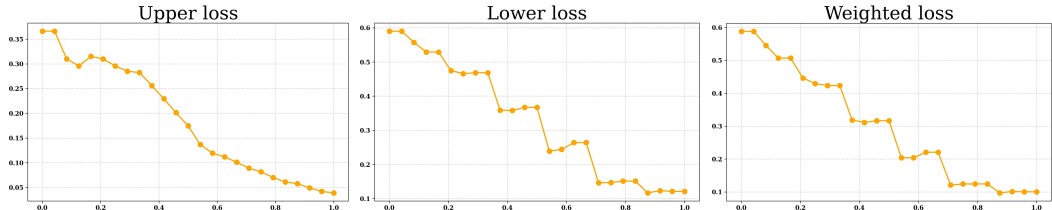

Figure 12: **Training loss curve of Instance Table — 10,000 samples.** The upper panel shows the convergence of the training loss, indicating stable optimization and steady improvement over time.

not align. In our setting, the instance table maintains one learnable weight per example, and with only 1,000 samples it can adapt extremely quickly; as a result, the inner loop effectively "outruns" the outer loop.

The weight-distribution snapshots in Fig. 11 further corroborate this diagnosis. Very early in training (10–30%), the histogram already collapses toward a narrow, peaked distribution, and by mid training the mass concentrates even more tightly. In other words, *weights converge too quickly*, saturating before the upper-level model has meaningfully adapted. This premature collapse reduces exploration over examples and decouples the inner updates from the outer objective: the upper-level loss continues to drift while the weights remain effectively frozen, producing the observed oscillations and the absence of steady improvement. Taken together, these results show that with an instance table on 1K samples, the inner weights overfit and settle far earlier than the outer model, yielding non-convergent bi-level behavior and a clear mismatch between the weight evolution and upper-level training dynamics.

**Instance Table — 10000 samples.**   Fig. 12 and Fig.13 show the training dynamics of the *Instance Table* variant when scaled to 10,000 samples. In contrast to the 1K case, all three loss curves (upper-level, lower-level, and weighted loss) display clear downward trends and converge smoothly. The oscillatory behavior observed with fewer samples is no longer present, suggesting that with a larger pool of instances, the optimization achieves a better balance between inner- and outer-loop updates. This indicates that the time-scale mismatch is greatly mitigated, and the bi-level optimization can proceed in a more synchronized fashion.

The weight distribution snapshots in Fig. 13 further highlight this improvement. Instead of collapsing prematurely, the learned weights remain well-spread and gradually evolve into stable, bell-shaped histograms across training stages. By the end of training, the distribution is concentrated yet not degenerate, suggesting that the model assigns meaningful and differentiated importance across examples without overfitting too early. Overall, with 10K samples the instance table exhibits both stable convergence in loss and healthy weight dynamics, demonstrating a much more desirable training regime compared to the 1K setting.

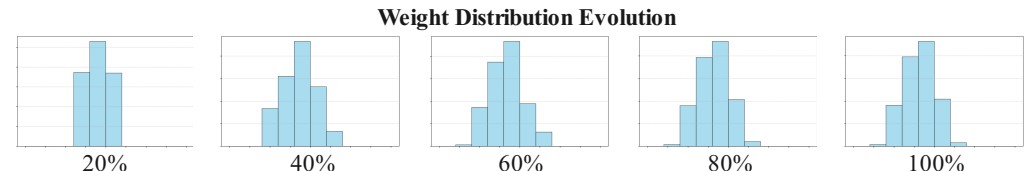

Figure 13: **Learned weight distributions of Instance Table — 10,000 samples.** The figure shows the distribution of learned domain weights at different training stages. The x-axis ranges from 0 to 2, with each bin corresponding to an interval of 0.2.

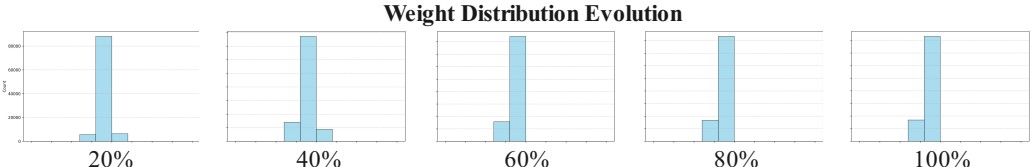

Figure 14: **Learned weight distributions of Instance Table — 100,000 samples.** The figure shows the distribution of learned domain weights at different training stages. The x-axis ranges from 0 to 2, with each bin corresponding to an interval of 0.2.

**Instance Table — 100000 samples.**   Fig. 14 illustrates the weight distribution evolution of the *Instance Table* trained with 100,000 samples. Unlike the 1K and 10K settings, the training here suffers from sluggish dynamics: the histograms remain narrowly peaked around their initialization and exhibit little change throughout the course of training. This indicates that the instance weights update extremely slowly when the parameterization scales to such a large dataset.

As a result, the optimization struggles to adjust importance across examples in a timely manner. The inner-loop learning is effectively "frozen," preventing the outer-loop PRM training from receiving meaningful gradients from the reweighting mechanism. This mismatch manifests as slow or stalled convergence, undermining the benefits of instance-level adaptation. In practice, with 100K samples the instance table loses much of its flexibility: instead of dynamically redistributing weights, the model maintains almost uniform assignments, which severely limits its effectiveness. Compared with the smoother convergence in the 10K case, the 100K setup highlights the challenge of scaling instance tables to very large sample sizes, where weight learning becomes a bottleneck.

**Instance Net — 1000 samples.**   Fig. 15 and 16 show the training dynamics of the *Instance Net* variant with 1,000 samples. In this low-data regime, convergence is unsatisfactory: although the upper-level loss decreases eventually, the lower-level loss fluctuates heavily and the weighted loss curve remains unstable, indicating inconsistent optimization signals between the two levels. This instability suggests that the network is unable to learn robust instance weights with such a limited number of examples.

The weight distribution snapshots in Fig. 16 further confirm this issue. Across all training stages, the histograms stay narrowly concentrated around their initialization with only minor variations. Unlike the broader, well-shaped distributions observed in larger-sample settings, here the weights exhibit almost no meaningful differentiation among instances. In other words, the Instance Net fails to effectively reweight examples, leaving the outer model without useful guidance. Overall, training with 1K samples leads to poor convergence and stagnant weight adaptation, highlighting the limitations of instance-level reweighting under very small data scales.

**Instance Net — 10000 samples.**   Fig. 17 and Fig.18 present the training curves and weight distribution dynamics of the *Instance Net* with 10,000 samples. On the surface, the loss curves appear well-behaved: both the upper-level and weighted loss steadily decrease, and the lower-level loss converges quickly without prolonged oscillations. This might suggest that optimization is stable. However, the weight distributions tell a different story. Across all training stages from 10% to 100%,

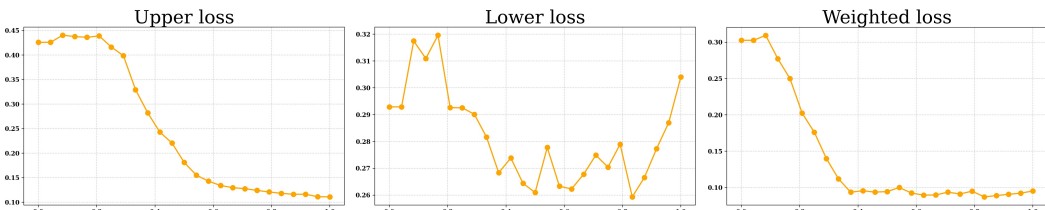

Figure 15: **Training loss curve of Instance Net — 1000 samples.** The upper panel shows the training loss.

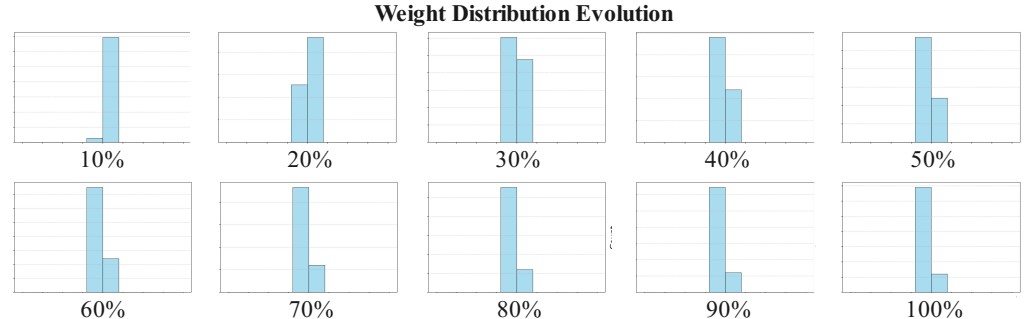

Figure 16: **Learned weight distributions of Instance Net — 1000 samples.** The figure shows the distribution of learned domain weights at different training stages. The x-axis ranges from 0 to 2, with each bin corresponding to an interval of 0.2.

the histograms remain nearly identical to initialization, with almost no visible spread or meaningful shift. In other words, the instance weights are effectively frozen and fail to adapt throughout training.

This phenomenon indicates that while the outer-level objective can still optimize with a large sample pool, the inner weighting mechanism ceases to play an active role: the network does not reassign importance across instances, and the reweighting module degenerates to a static mapping. As a result, the *Instance Net* at 10K samples provides little benefit over a uniform baseline, and the lack of dynamic weight adjustment renders this setting uninformative. For this reason, we excluded the 10K configuration from further analysis and focused instead on regimes where weight adaptation exhibited meaningful evolution.

**Instance Net — 100000 samples.** Fig. 19 shows the weight distribution evolution of the *Instance Net* trained with 100,000 samples. Compared to the 1K and 10K settings, this configuration exhibits by far the most desirable behavior. The distributions start close to uniform but progressively reshape over time: by 30–50% of training, the histograms spread across the full range, indicating that the model begins to differentiate strongly between easy and hard examples. As training progresses toward 80–100%, the weights stabilize into a well-structured distribution with a clear long-tail, demonstrating that the reweighting module has successfully learned to assign heterogeneous importance across instances.

This result suggests that at large scale, the instance net overcomes the limitations seen in smaller sample sizes: it avoids premature collapse (as in 1K) and bypasses the stagnation where weights barely move (as in 10K). Instead, with 100K samples the weight adaptation is both dynamic and stable, offering the strongest evidence that instance-level reweighting works as intended when sufficient data are available. Accordingly, the 100K setting represents the most effective balance for *Instance Net*, showing that scaling up not only improves convergence but also yields meaningful and interpretable weight dynamics.

# H MORE DISCUSSIONS ON PARAMETER NUMBER

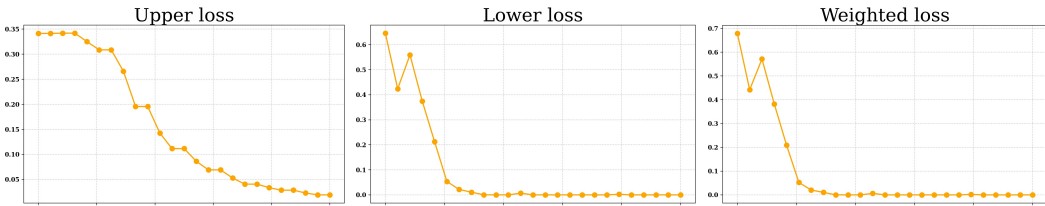

Figure 17: **Training loss curve of Instance Net — 10000 samples.** The upper panel shows the convergence of the training loss, indicating stable optimization and steady improvement over time.

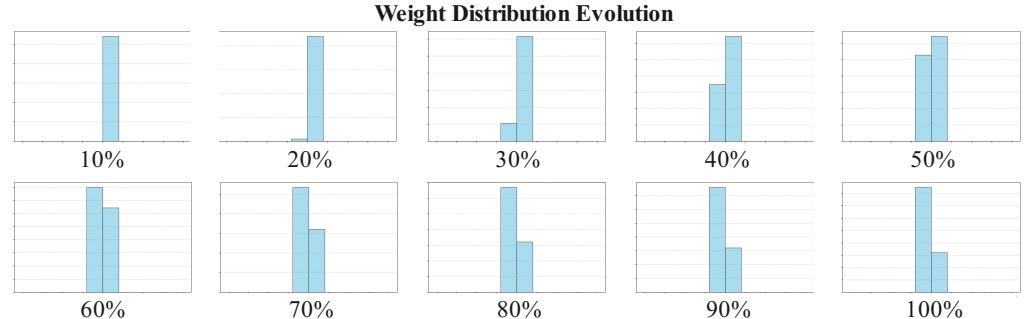

Figure 18: **Learned weight distributions of Instance Net — 10000 samples.** The figure shows the distribution of learned domain weights at different training stages. The x-axis ranges from 0 to 2, with each bin corresponding to an interval of 0.2.

**Convergence curve.** Fig. 20 compares the upper-level loss trajectories under different parameterizations. We observe that the number of parameters has only a minor effect on the convergence *speed*: all settings exhibit a smooth and consistent decrease of the upper loss across training, reaching their stable phase within a comparable number of iterations. However, the parameter scale substantially influences the *final convergence value*. For instance, the Domain Table variant settles at a higher plateau around 0.15, whereas the other parameterizations (Instance Table, Instance Net) converge much lower, around 0.05. This indicates that although increasing or decreasing parameter count does not significantly alter how quickly optimization proceeds, it plays a decisive role in determining the eventual optimum reached. In practice, larger or more expressive models achieve better minima, highlighting that parameterization mainly impacts the quality of the converged solution rather than the dynamics of getting there.

**Instance Net — hidden dim = 30.** Fig. 21 shows the weight distribution evolution of the *Instance Net* with hidden dimension set to 30. The results are qualitatively very similar to those obtained with a smaller hidden size (e.g., hidden dim = 10). Across training stages, the histograms evolve in the same pattern, quickly moving from near-uniform initialization toward a skewed distribution, and then stabilizing without substantial differences compared to the lower-dimensional setting. This suggests that increasing the hidden dimension does not yield additional representational benefits for the weighting function. In other words, the complexity introduced by a larger hidden dimension does not translate into more meaningful or diverse weight adaptation. Hence, there is little motivation to increase the hidden size beyond 10, as the results remain almost unchanged while incurring higher computational cost.

**Instance Net — hidden dim = 50.** Fig. 22 presents the weight distribution dynamics of the *Instance Net* with hidden dimension increased to 50. Compared with smaller hidden sizes (e.g., 10 or 30), the distributions evolve much more slowly: even at later stages of training (60–80%), the histograms still retain patterns close to initialization, and only by the very end (90–100%) do they start to diverge more noticeably. This indicates that a larger hidden dimension significantly slows down convergence, likely because the weight prediction network becomes over-parameterized relative to the available signal.

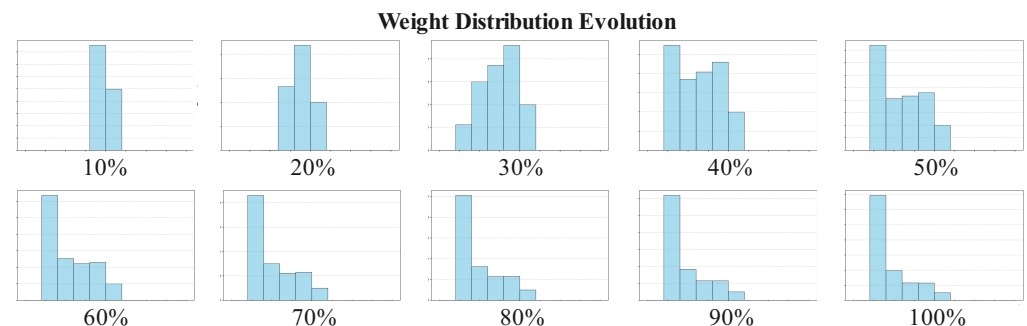

Figure 19: **Learned weight distributions of Instance Net — 100,000 samples.** The figure shows the distribution of learned domain weights at different training stages. The x-axis ranges from 0 to 2, with each bin corresponding to an interval of 0.2.

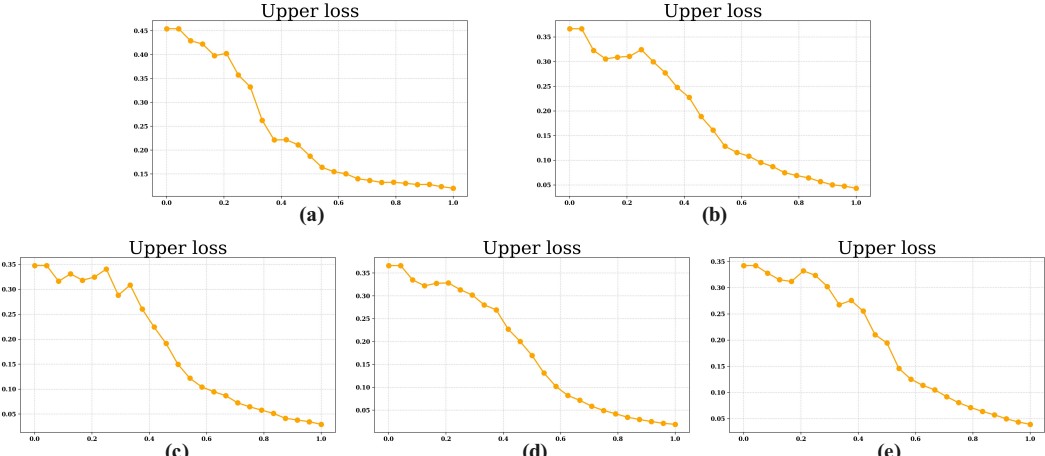

Figure 20: **Upper loss curve of parameter test.** (a) Domain Table. (b) Instance Table. (c) Instance Net (hidden dim = 10). (d) Instance Net (hidden dim = 30). (e) Instance Net (hidden dim = 50).

In practice, although higher capacity could in theory provide more expressive reweighting, our observations show that it instead delays adaptation and hinders efficient optimization. Thus, increasing the hidden dimension to 50 does not yield practical benefits and, on the contrary, makes training less efficient. A smaller hidden size (10 or 30) is already sufficient for capturing the relevant weighting patterns while converging much faster.

**Domain Table.** Fig. 23 illustrates the evolution of learned weights for the *Domain Table*. Compared to instance-level methods, the domain-level parameterization provides only a limited number of learnable weights, each tied to an entire domain rather than individual examples. As a result, the histograms across training stages show relatively coarse and rigid distributions: although there is some variability and fluctuation, the number of bins populated remains small, and the shape does not exhibit the finer-grained adaptation seen in instance-based approaches.

This behavior indicates that the Domain Table lacks sufficient flexibility to fully exploit the heterogeneity of the training set. With too few weights, the model cannot meaningfully distinguish among samples within the same domain, leading to under-utilization of the reweighting mechanism. In practice, this constraint manifests as limited expressiveness and suboptimal convergence compared with Instance Table or Instance Net, which can allocate weights more granularly and dynamically.

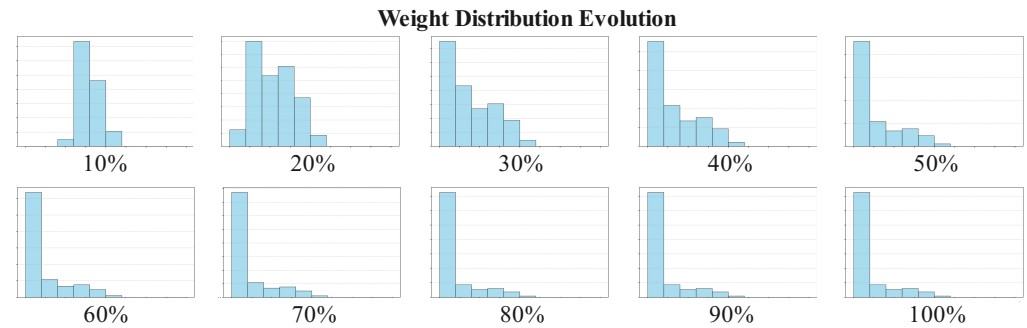

Figure 21: **Learned weight distributions of Instance Net — hidden dim = 30.** The figure shows the distribution of learned domain weights at different training stages. The x-axis ranges from 0 to 2, with each bin corresponding to an interval of 0.2.

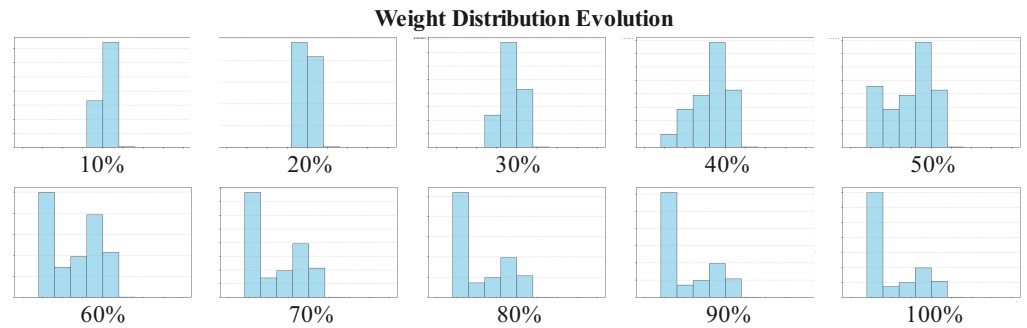

Figure 22: **Learned weight distributions of Instance Net — hidden dim = 50.** The figure shows the distribution of learned domain weights at different training stages. The x-axis ranges from 0 to 2, with each bin corresponding to an interval of 0.2.

## I ABLATION STUDY DETAILS

Table 5 reports an ablation study on two critical training strategies: (i) removing the cold-start initialization (*w/o cold start*), and (ii) removing the + and - aggregation loss (*w/o both loss*). The results highlight how these design choices differently affect the Instance Table and Instance Net variants.

For the Instance Table (trained with 1000 samples), both ablations cause noticeable degradation. Without cold-start initialization, accuracy drops from 83.8 to 82.2 (–1.6), and sanity check performance decreases from 88.7 to 88.0 (–0.7). This indicates that the Instance Table is sensitive to initialization: starting from uniform or unstable weights makes it harder for the model to quickly differentiate examples, leading to suboptimal convergence. The effect is even stronger when removing the coupled losses: accuracy decreases by –1.2, and the sanity check metric collapses by –3.5, showing that bi-level optimization is essential for properly aligning instance-level weights with the outer PRM objective. In other words, Instance Table relies heavily on carefully designed training signals to avoid degenerate solutions.

By contrast, the Instance Net (hidden dimension = 10, trained with 100000 samples) demonstrates greater robustness. When trained without cold start, its accuracy also drops (–1.7), but its sanity check *improves* slightly (+1.0). Similarly, removing both losses produces only a marginal change in accuracy (–0.3) while again increasing sanity check performance (+1.3). These results suggest that Instance Net, due to its parameterized weight prediction network, is less sensitive to initialization and can still learn useful weighting dynamics even when explicit bi-level losses are absent. The improvements in sanity check imply that the network's inductive bias promotes smoother and more consistent weighting, even if the accuracy metric does not fully benefit.

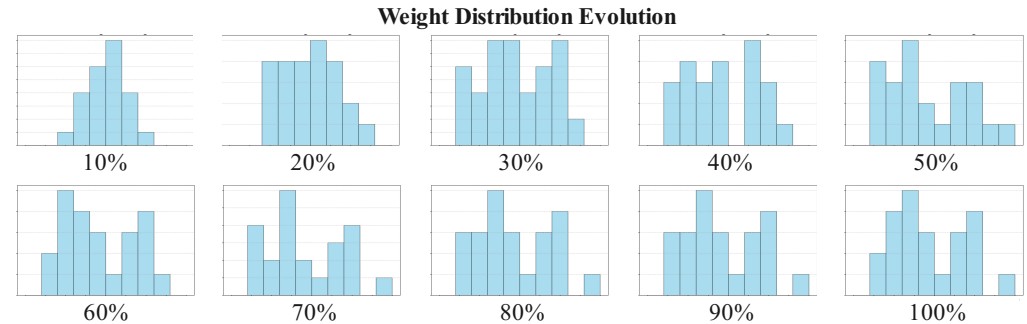

Figure 23: **Learned weight distributions of Domain Table.** The figure shows the distribution of learned domain weights at different training stages. The x-axis ranges from 0 to 2, with each bin corresponding to an interval of 0.2.

Taken together, the ablation study reveals complementary characteristics: *Instance Table* can yield strong performance but requires careful initialization and tightly coupled optimization objectives to avoid instability, while *Instance Net* offers more robust and self-regularizing behavior at the cost of slightly weaker accuracy. These findings justify our combined exploration of both designs: Instance Table highlights the potential upper bound when supervision is well-structured, whereas Instance Net shows resilience under relaxed training regimes.

## J   THE USE OF LARGE LANGUAGE MODELS

We used large language models (LLMs) solely as a general-purpose writing assistant. Specifically, LLMs were employed to help rephrase sentences, check grammar, and polish the presentation of the paper. LLMs did not contribute to research ideation, experimental design, implementation, or analysis. All scientific content, methodology, and results reported in this paper are entirely the responsibility of the authors. The authors retain full accountability for the correctness and originality of the work.

