# OpenReview forum: "OraclePRM: Unlocking the Potential of Each Instance for Multimodal Process Reward Model Training"
_ICLR.cc/2026/Conference — Submitted to ICLR 2026_

### Official Review · Reviewer_cb3T · 2025-10-14

**Soundness:** 2
**Presentation:** 2
**Contribution:** 2
**Rating:** 2
**Confidence:** 2

**Summary:**

This paper proposes **OraclePRM**, an instance-level reweighting framework for multimodal Process Reward Model (PRM) training. The method formulates per-example weights via a bi-level objective and provides two practical instantiations: Instance Table (explicit per-sample weights; suited to small/medium data) and  Instance Net (a lightweight network that generalizes and scales to large corpora).  Reported results include strong performance on MMMU (e.g., 84.6) and first-place leaderboard standings when paired with a leading backbone (e.g., GPT-5-mini), suggesting the approach narrows the gap toward an oracle upper bound probed via a “sanity check.”

**Strengths:**

**Clear move from domain to instance-level reweighting.** The paper motivates why domain-level reweighting (e.g., DreamPRM) can leave headroom and positions OraclePRM as a principled instance-wise alternative that better matches test-time objectives.


**Two scalable instantiations.** The split between *Instance Table* and *Instance Net* offers a practical continuum across data regimes, making the approach adoptable in different scales and settings.

**Weaknesses:**

**Unclear practical significance and breadth of application.** The paper does not yet make it concrete where OraclePRM decisively outperforms simpler data-curration or domain-weighting in real applications.

**Limited backbone diversity in the primary figure.** If Figure 3 primarily uses GPT-5-mini as the backend, that is a narrow setting. Results on several mainstream LLMs (e.g., GPT-4o, GPT-5) would strengthen the claim that OraclePRM is backbone-agnostic and robust.

**Convergence and generalization remain unclear.** The bi-level optimization is central, but desirable convergence properties are not established. Figures 4 and 6 show training loss (a limited signal); furthermore, if Figure 4’s training loss reaches (near) zero, this raises overfitting concerns.

**Questions:**

**Uncertainty  significance.** Can you report mean and 95% CI (or bootstrap CIs) for all headline metrics in Figure 3 and Table 1? And can you match the 8-run protocol from Table 7 for  OraclePRM results so it is easiser to determine whether the lift is stats significant?


**Optimization properties.** Do you have theoretical guarantees or ablations  that speak to convergence/stability of the bi-level updates?

---

### Official Review · Reviewer_CuUd · 2025-10-27

**Soundness:** 2
**Presentation:** 2
**Contribution:** 3
**Rating:** 4
**Confidence:** 3

**Summary:**

The paper proposes OraclePRM, an instance-level reweighting framework for training multimodal Process Reward Models (PRMs) via bi-level optimization. To scale across data regimes, the authors introduce two complementary instantiations: InstanceTable and InstanceNet. To stablize the training, this work proposed to utilize time-scale matching between upper (meta) and lower (PRM) updates, cold-start initialization, and bounded weight ranges (clip/sigmoid). The upper level learns weights using a meta-loss (MSE) on an aggregated trajectory score that emulates test-time scaling (Best-of-N), while the lower level trains the PRM with instance-weighted step-wise cross-entropy. Experiments show strong gains on MMMU and several multimodal math benchmarks.

**Strengths:**

1. The paper is generally well-written and easy to follow, with a clear description of the method.
2. The paper provides intuitive visual demonstrations to help better understand the paper.

**Weaknesses:**

1. **InstanceNet feature design**. Why use only the final PRM hidden representation $h(x)$? Did you try early/mid-layer features, vision-only embeddings, or multimodal cross-attention outputs? Any results on feature ablations and their impact on generalization (Table 1) vs. sanity check?

2. **Beyond math benchmarks.** Given Figure 8’s motivation (modality necessity varies), can you report evaluations on non-math, image-dependent benchmarks (e.g., chart/diagram QA, scientific figures, medical imagery) and noisy OCR scenarios to validate robustness?

3. **Capacity vs. overfitting for Instance Table.** With large datasets (100k), Instance Table’s weights barely move (Figure 14). Would structured parameterization (e.g., hashing, per-bucket/table-lookup by difficulty/modality), weight decay on $\alphaα$, or EMA updates alleviate stagnation and improve convergence?

4. **Meta-loss design and selection quality.** Can you compare MSE on mean-aggregated step scores to ranking losses (pairwise/softmax list-wise) and to calibration-aware objectives? Do these reduce overfitting or improve selection under Best-of-N? Any results with alternative aggregators (e.g., max, top-k mean, attention-weighted) and per-step consistency regularization?

**Questions:**

See the `Weaknesses` part.

---

### Official Review · Reviewer_XrRs · 2025-10-31

**Soundness:** 2
**Presentation:** 3
**Contribution:** 2
**Rating:** 4
**Confidence:** 4

**Summary:**

This paper introduces OraclePRM, an instance-level reweighting framework for training multimodal process reward models. Unlike prior domain-level methods, OraclePRM assigns adaptive weights to individual training samples through bi-level optimization, allowing the model to better handle data quality imbalance and distribution shift. The method incorporates stability mechanisms including time-scale matching, cold-start initialization, and bounded weights, and supports two regimes: Instance Table for explicit per-sample weighting and Instance Net for scalable generalization. Overall, OraclePRM provides a stable and effective approach to improving supervision fidelity and generalization in multimodal reasoning.

**Strengths:**

- Overall, the paper is clearly written and easy to follow.
- Experiments across multiple base models and multimodal benchmarks demonstrate the effectiveness of the proposed framework.
- The authors provide additional analyses, such as weight distribution and sanity checks, to support the validity of the proposed approach.

**Weaknesses:**

- Limited techical contribution. The proposed method is heavily dependent on the prior work DreamPRM. Data reweighting, bi-level optimization, and cold-start initialization are well-established techniques in multimodal learning, and the paper primarily combines them in the context of multimodal process reward modeling.
- Limited analysis of the adaptive weighting behavior. For the instance-level weighting, what types of data tend to receive higher weights at different stages of training? It would be helpful if the authors could provide additional analysis or qualitative examples to validate the effectiveness and interpretability of the learned weights.
- The evaluation scope is narrow. It would strengthen the paper to include results on standard reward modeling benchmarks (e.g., VL-RewardBench, VisualProcessBench) to directly assess the quality of the proposed PRMs.

**Questions:**

Please refer to the weaknesses section for details.

---

### Official Review · Reviewer_226y · 2025-11-01

**Soundness:** 2
**Presentation:** 2
**Contribution:** 2
**Rating:** 4
**Confidence:** 4

**Summary:**

This paper aims to address the limitation of Multimodal Process Reward Models (PRMs), which scores reasoning steps (not just final answers) for tasks where both images and text are involved. But training them is difficult due to two core issues: 1.Data quality imbalance — training datasets contain many trivial, noisy, or unhelpful examples; 2.Distribution shift — the data used for training differs significantly from what models face during test-time reasoning.

Existing method DreamPRM performs domain-level re-weighting (assigning one weight per dataset) to mitigate this. However, it still falls short of the theoretical upper bound (oracle performance) because one weight per dataset is too coarse and cannot capture differences within datasets. The paper proposes OraclePRM, which assigns a separate weight to every training instance, making training more sensitive to differences in example quality.  This is done using bi-level optimization: 1. Lower level: Train PRM with instance-weighted loss;
2. Upper level: Adjust instance weights using validation/meta performance so that weighting aligns with the target task.

**Strengths:**

The proposed OraclePRM framework introduces instance-level bi-level re-weighting that directly optimizes sample importance with respect to a meta-objective aligned to test-time selection. The method is intuitive yet non-trivial, and connects naturally to meta-learning and data curation literature, while providing a clean extension beyond prior domain re-weighting approaches like DreamPRM.

Experiments contain multiple multimodal reasoning benchmarks and PRM backbones. The method consistently outperforms baseline PRMs (SFT, verifier-style, DreamPRM), and in controlled settings, approaches the oracle upper bound for Best-of-N decoding. The results demonstrate both robustness and practical significance.

**Weaknesses:**

The proposed OraclePRM relies on the held-out meta dataset whose distribution is aligned with the target test-time tasks. In practice, constructing such a representative meta dataset might be non-trivial.

The bi-level re-weighting introduces the training stability challenges. The paper has introduced several techniques such as cold-start initialization, weight clipping, and update-time-scale matching etc. to avoid divergence. these techniques may require exhaustive tuning and could reduce robustness across model or dataset scales.

**Questions:**

1. What's the "threshold" of dataset size to transition from "Instance Table" to "Instance Net"? What's the intuition behind that? What are the training cost for both Instance Table and Instance Net?

2. What's the training strategy if we don't have the representative meta data? Or if the distribution of meta data does not align with the test-time tasks, what should we do?

3. If we do any data augmentation, do we need to re-train either the Instance Table or Instance Net?

---

### Meta-Review · Area_Chair_7AtT · 2026-01-07

**Summary:**

The AC carefully read the paper and the full discussion. The submission received scores consistently below the acceptance threshold (scores: 4, 4, 4, 2). Reviewers generally agreed that the paper targets an interesting problem—improving Multimodal Process Reward Models (PRMs) by moving from domain-level to instance-level reweighting—and appreciated the intuitive motivation and strong results on math benchmarks like MMMU.

However, the primary concerns center on the practical difficulty of the meta-learning setup (Reviewer 226y), limited technical novelty and narrow evaluation (Reviewer XrRs), lack of transparency regarding feature design and alternative benchmarks (Reviewer CuUd), and concerns over statistical significance and optimization stability (Reviewer cb3T).

The authors didn't provide the rebuttal and all reviews are negative so the AC recommends to the rejection.

**Reviewer Concerns:**

No rebuttal provided so no concerns were addressed.

**Reviewer Scores:**

No rebuttal provided so no scores changed.

---

### Decision · Program_Chairs · 2026-01-26

Reject